# Attitudes and Involvement of Employees in the Process of Implementing Innovations and Changes in Companies

**DOI:** 10.3390/bs12060174

**Published:** 2022-06-01

**Authors:** Zuzana Lušňáková, Renáta Benda-Prokeinová, Zuzana Juríčková

**Affiliations:** Faculty of Econonomics and Management, Slovak University of Agriculture in Nitra, Tr. A. Hlinku 2, 94976 Nitra, Slovakia; renata.prokeinova@uniag.sk (R.B.-P.); zuzana.jurickova@uniag.sk (Z.J.)

**Keywords:** human resources, perception and acceptance, text mining

## Abstract

The main goal of the paper is to evaluate the perception and acceptance of change by human resources in companies in Slovakia and to propose specific recommendations to increase the involvement and active approach of business representatives in the interest of business sustainability. Based on the set goal, the issue of perception and attitudes of employees were evaluated through data obtained by a questionnaire survey through a sample survey in 816 companies operating in the Slovak Republic. The obtained data were analyzed using association analysis by application of the independence test based on formulated hypotheses and the found dependencies were presented by correspondence analysis. Innovative activities of companies do not always have to be perceived only positively by employees and it is also appropriate to consider the opposite behavior of employees. To minimize adverse events, we recommend companies formulate appropriate strategies to reduce and overcome the effects of negative attitudes to organizational change and then implement them appropriately, which is essential for the successful implementation of innovative practices.

## 1. Introduction

Implementing innovation and innovative practices in business activities is a global topic. Companies that follow current trends, implement innovative procedures, and invest in quality inputs, meet the basic prerequisites to be successful and sustainable companies in the future. The most important and irreplaceable factor in companies are people. It is their perception, commitment, loyalty, and diligence, but also knowledge, skills, and experience that are the basic prerequisites for the successful implementation of new procedures and processes to streamline work procedures, gain a competitive advantage, and ensure sustainability. Based on the above, the main goal of the paper is to evaluate the perception and acceptance of change by human resources in companies and to propose specific recommendations to increase the involvement and active approach of business representatives.

Organizations are seeking to increase their competitive advantages, taking more market, more customers, and more sales. Rapid changes stemming from globalization, advancement of information systems, and other factors have caused higher competition. The realization of goals will be achieved through the human resources management in organizations. The workforce, as the key to success, will enable the achievement of organizational performance [1]. The competitive advantage of companies could also be connected to the sustainability of the work force, decreased fluctuation of employees, and necessary training, with a positive impact on the whole efficiency of the company [2].

Human Resource Management (HRM) plays a key role in the four widely accepted profit-making mechanisms, although not always as predicted by mainstream strategic human resource management. HRM performance studies based on a company’s resource-based view always focus on human resources that the company already controls. Specifically, these are resources that are scarce, inimitable, and irreplaceable and can be used throughout the organization [3].

Both the strategy and Human Resources Management (HRM) types of literature recognize the importance of human capital for enhancing firm performance. People’s behaviors also need to be in line with the strategic goals to add value to the organization, and purpose and identity are important constructs to include when explaining human capital processes and effectiveness [4]. Organizations need to manage their human resources effectively and efficiently to achieve the desired goals and objectives [1]. For example, the findings of [5] provide an overview of investment levels where a human resources management system can have a positive impact on a company’s performance.

Over the last 30 years, strategic human resource management has become the dominant approach to human resource management policy [6]. It has a clear focus on implementing strategic change and growing the skill base of the organization to ensure that the organization can compete effectively in the future [7].

Change is unavoidable in the modern business world and it needs to be embraced fully and implemented inclusively for all its benefits to be realized. Senior management needs to create a coherent overview of the change they are seeking and engage their staff in a consistent, meaningful, and rewarding manner: reward success and tolerate failure [8].

Organizational change is essential for short-term competitiveness and long-term survival, but it poses daunting managerial challenges [9]. Managing change and organizational development has emerged as effective tools to survive in the fast-paced business environment. Organizations invest a significant amount of resources and energy in trying to change and for the change initiative to be successful; organizational members must engage in the change program and remain committed to it. Therefore, many organizations, in today’s dynamic environment, are considering and exploring the concept of change management and employee engagement [10].

Commitment to change reached its highest level when change demands occurred primarily at the unit level, change demands at the individual level were low, and change was deemed favorable. If change was seen as generally unfavorable, commitment dropped [11]. Changes in the form of organizational innovations have several manifestations and links, to which it is necessary to respond appropriately and choose appropriate procedures for their management. Every change requires considerable attention, as it has various consequences. According to Kožárová [12], this is not a simple issue. It requires a perfect knowledge of the company, a differentiated approach, managerial skills, and abilities. It depends on how human resource management professionals view organizational change and its roles, as this perception serves as the basis for how they define their roles. Human resources professionals hold numerous roles in change efforts, including those of “change agent” and “consultant.” Additionally, most human resource management professionals tended to view successful organizational change as primarily occurring in a top-down, hierarchical manner [13]. The basic determinants in the process of implementing innovative procedures and changes are employees. It is extremely important to perceive how changes affect them, and whether they can deal with them and accept them. According to findings of Yue et al. [14], transformational leadership and transparent communication were positively associated with employee organizational trust, which in turn, positively influenced employee openness to change.

Human resources, as the most important factor of production, transforms finances and materials into required products or services and brings companies additional material and financial resources. Human resources need to be competent to be able to perform tasks related to dynamic and turbulent times at the required professional level. These facts require management to focus on a high degree of involvement and participation of employees, who have an important voice in decision-making [15].

Employee involvement is important for organizations because it increases employee engagement and participation, which in turn increases the organization’s competitiveness [16]. The very involvement of employees has a positive impact on the individual as well as on the results of the organization. The person–organization relationships positively contributed to both work engagement and organizational engagement [17]. However, we found that the contribution of the person–organization fit to organizational engagement was more powerful than work engagement. A positive work culture enhances employee engagement and in specific cases leads to increased adaptability [18,19]. Whereas organizational engagement has a positive effect on the individual’s ability to adapt to changes, job engagement has the opposite effect, uncovering potential obstacles to change management in organizations. Valence and trust in leadership individually and sequentially also mediate the relationship between transformational leadership and employee engagement [20].

Mazzanti, Pini, and Tortia [21] also emphasize the human resources aspect and point out that new practices initiated by managers could be more effective if employees are actively involved. The concept of innovation with high involvement of all employees in innovation positively affects the economic results of the company. However, there is another positive effect. The more people are involved in change, the more willing they are to accept change or implement it themselves [22]. Employees, therefore, need to be supported in their innovative efforts.

Engagement has a significantly positive impact on productivity, performance, and organizational advocacy, as well as individual wellbeing, and a significantly negative impact on intent to quit and absenteeism from the workplace [23].

Managers must communicate their understandings, particularly during organizational change, in a way that provides their subordinates with a workable certainty. Consequently, “sensemaking” becomes very important for engagement to achieve and manage change successfully [9].

Motivation and the way of communication and cooperation can be suitable tools, which create space for employees to self-realize. These can all be a sufficient incentive to engage in the progress of society [24]. Swarnalatha and Prasanna [11] see also strong communication, collaboration, information flow, trust, and effective problem solving as the common key functions of engagement and change. If employee engagement is a primary antecedent to successfully implementing an organizational change initiative, then deficiencies in these key functions form a potential a barrier to employee engagement as well as change initiative.

Lenberg et al. [25] also wrote about the employees’ attitudes to organizational change and the implementation of innovative practices, because they are critical determinants in the process of change. Therefore, he tried to determine which basic concepts affect them. They identified three potentially important basic concepts:-knowledge of managers about the intended result of the change;-their understanding of the need for change;-feelings of participation in the process of change.

Negative employee behavior must also be considered when implementing innovations. In order to prevent undesirable behavior (aversion or resistance), companies can choose appropriate strategies to reduce and overcome the negative effects of resistance to organizational change and subsequent proper implementation of these strategies, which are necessary for successful implementation of organizational change [26]. Research conducted by Erwin and Garman [27] provides managers and implementers of organizational change with a practical guide to understanding and managing employees’ negative attitudes toward change.

The issue of organizational innovation and related concepts in the field of human resource management aimed at overcoming employees’ resistance to change is extremely important and current because change is becoming an everyday reality with which managers are confronted [12].

Perceiving and seizing opportunities to improve employee competence, service quality, quality improvement programs, or relationships between organizational employees would also contribute to preparedness for the change process. A survey by Vakola and Nikolaou [28] on employees’ readiness for change shows that their perception of the risks of overworking also affects their readiness and attitude to organizational change. For the success of organizational change, the mental state of individuals must also be taken into account. Employee participation in the process of organizational change is essential and the skills of leaders are very important for this process. Leaders must also be able to convince employees who resist change. This is another role of leaders in the process of change [29]. Employees can also express a distrustful attitude towards newly hired external employees, especially if they hold the positions of team leaders, projects, etc. [30]. They see competitors in them and are worried about their careers. These attitudes of existing employees can be overcome only by continuous work, focused on communication and explaining the benefits of open innovation, as well as by looking for effective forms of motivating people.

In this context, it is important to encourage employees to exchange knowledge and build their relationships with the company, as well as to support the idea of exchanging information and knowledge between employees, organize frequent meetings, and provide appropriate infrastructure for meetings, including virtual ones [15].

Although much attention has been paid to understanding and defining employees’ resilience to change, relatively little research examines the impact of employee positivity on organizational change. To help meet this need, Avey et al. [31] examine whether the process of employee positivity will influence relevant attitudes and behaviors. Authors found out that:-their psychological capital (a basic factor consisting of hope, efficiency, optimism, and resilience) was related to their positive emotions, which in turn were related to their attitudes (engagement and cynicism) and behavior (organizational affiliation and limitations) related to organizational change;-mindfulness (i.e., increased awareness) was related to psychological capital in predicting positive emotions;-positive emotions generally mediate the relationship between psychological capital and attitudes and behaviors.

In addition, the findings of Soni and Rastogi [32] suggested psychological capital as a significant predictor, along with the other variables in fostering employee engagement.

Based on the above, three research assumptions were formulated:RA1: We assume that companies that follow a strategic approach to human resources to support innovative practices also implement innovative practices in the field of human resources.RA2: We assume that companies implementing innovative practices in the field of human resources also do everything they can to make employees feel engaged and dedicated to their work.RA3: We assume that in companies whose management is interested in the opinions and attitudes of employees on the issue, according to managers, changes and innovations are perceived positively by employees.

## 2. Methodology

The main goal of the paper is to evaluate the perception and acceptance of innovative practices and changes by human resources in companies and to propose specific recommendations to increase the involvement and proactive approach of business representatives in the interest of business sustainability. Based on the set goal, the issue of perception and attitudes of employees have been evaluated through data obtained by a questionnaire survey through a sample survey in companies operating in the Slovak Republic. Based on the theoretical basis, a questionnaire was formulated as a tool for obtaining information from business practice. A total of 816 companies (out of 1240 asked) were randomly involved in the survey, which was conducted from May 2020 to May 2021.

Specific questions (Q1–Q5) concerning the addressed issue of “perception and acceptance of the implementation of innovative procedures in human resource management in companies in Slovakia” were formulated as positive statements.

Q1: Company follows a strategic approach to human resources to support innovative practices.Q2: The company implements innovative practices in the field of human resources.Q3: Company is interested in the opinions and attitudes of their employees.Q4: The management of the company does everything to make employees feel engaged and dedicated to their work.Q5: From the management of the company’s point of view, changes and innovations are perceived positively by employees.

Within each positive statement, the task of the respondents was to express agreement or disagreement on a 5-point Likert scale and to specify their answers verbally. As part of the analysis of respondents’ answers and presentation of research results, we also used numerical designation of answer options, where the number 1 represented absolute disagreement with the statement, 2 partial disagreement, 3 indeterminate answer, 4 partial agreement, and 5 absolute agreement with the statement. In connection with the interpretation of the findings resulting from the analysis of data, the higher the average value achieved, the more positive and consistent the respondents’ attitude to the issue will be. The survey respondent was a human resources manager or another manager who was also responsible for the human resources department.

To verify the representativeness of the sample, we applied the Chi-square test of the good agreement for all identification questions. The sample is representative of the identifiers “Business size”, “Industry,” and “Capital participation of the company”. To verify the reliability of the questionnaire, Cronbach’s Alpha coefficient was applied [33]. In connection with validation, the Fleiss Kappa coefficient κ was applied [34,35]. Due to the structure and nature of the obtained data, non-parametric methods were used in their statistical processing. An association analysis was applied to determine the dependence between the answers to the questions of the questionnaire survey, focused on the perception and acceptance of the implementation of innovative procedures in working with human resources. The Cramer V contingency coefficient was determined by applying the chi-square test of independence [36,37]. This, in turn, made it possible to interpret the relationship between the two questions in the questionnaire. After identifying the dependencies, we examined the internal structure of contingency tables through correspondence analysis, where it is necessary to categorize continuous variables [38]. The application of correspondence analysis is therefore preceded by testing the hypothesis of the independence of the observed features in the contingency table. Correspondence analysis makes it possible to examine the association of categorical variables [39] and to obtain a clear graphical representation of the context in two-dimensional resp. multidimensional space. The aim is to assess the interrelationship between the variables and to explain the structure of the examined dependence.

From the point of view of the practical application of this method, it would be appropriate for us to be able to visually identify the relationships between the rows and columns of the contingency table. Of course, this is not possible in a multidimensional space. Therefore, it is necessary to find the projection of the points of multidimensional space representing the rows and columns of the contingency table into the plane, in order to obtain the so-called correspondence map, which we then interpret relatively easily. A mosaic plot (Marimekko diagram) is a graphical method for visualizing data from two or more qualitative variables [39]. The first step in the analysis was to determine the most common and average response (mode and average) in the questionnaire survey. It gives an overview of the data and makes it possible to recognize relationships between different variables. Independence is presented if the boxes across categories all have the same areas. The closer the points in the correspondence map are, the similar the categories are and the more they correspond to each other. In some cases, in the case of so-called symmetric correspondence maps, we can sometimes use the position of the points relative to the main axes of the correspondence map in the interpretation.

## 3. Results

We examined the perception and acceptance of the implementation of innovative procedures in working with human resources in companies in Slovakia from the perspective of company management. We were interested in whether the competent people were interested in the opinions and attitudes of their subordinates, and how, according to the management, employees perceive these procedures, and whether the company does everything for the satisfaction and dedication of its employees.

The introduction of the questionnaire consisted of five identification features (sorting items) of respondents: the size of the enterprise (small, medium, large), legal form of an enterprise (joint-stock company, limited liability company, cooperative, other), capital participation of enterprise (exclusively domestic enterprise, exclusively foreign enterprise, and enterprise with combined capital participation), branches of the national economy (primary, secondary, tertiary and quaternary), and the region in which the enterprise is located.

The research sample is listed in Table 1 and broken down by identification criteria.

In order to examine the perception, attitudes, and involvement of employees in the process of implementing changes and organizational innovations, data from a questionnaire survey were identified, processed, and analyzed.

The first step in the analysis was to determine the most common and average response (mode and average) in the questionnaire survey. The results of this part of the analysis show that up to 90% of respondents from companies partially or completely confirmed the fact that their company deals with the views and attitudes of employees, to the implementation of innovative procedures in working with human resources, and according to 69% of respondents, managers and employees perceive the implementation of innovative procedures in the work with human resources positively.

Based on the theoretical basis, three research assumptions were set. Their acceptance or rejection was made based on verification of statistical hypotheses through the application of the Chi-square test of the square contingency. The results are presented in Table 2.

By formulating the first research assumption, we investigated whether there is a positive relationship between the strategic approach of companies in the field of human resources in order to support innovative practices and the implementation of innovative practices in the field of human resources.

RA1: We assume that companies that follow a strategic approach to human resources in order to support innovative practices also implement innovative practices in the field of human resources.

**Hypothesis** **0** **(H0).**
*There doesn’t exist a dependence between the strategic approach of companies in the field of human resources in order to support innovative practices and the implementation of innovative practices in the field of human resources.*


**Hypothesis** **1** **(H1).**
*There exists a dependence between the strategic approach of companies in the field of human resources in order to support innovative practices and the implementation of innovative practices in the field of human resources.*


The results obtained by the χ^2^ test document the values in the first row of the previous Table 2. Based on the observed value of probability, which reached a value of less than 0.05, we can state that the existence of a moderately statistically significant dependence was confirmed, as the value of the correlation Cramer V coefficient is 0.5786 [40,41]. We state that companies that follow a strategic approach to human resources in order to support innovative practices also implement innovative practices in the field of human resources.

Based on the confirmation of the dependence between the answers to the first and second questions, we subsequently analyzed the relationship more deeply through correspondence analysis. In the correspondence map, we can interpret the relative position of the categories within one variable, and at the same time their position relative to the categories of the other variable. We measure the degree of similarity based on Euclidean distances between points within one variable. Those respondents who chose Preference 5 in the first question most likely also answered by selecting Preference 5 in the second question. The answers to both questions are consistent. Respondents’ answers were reflected in a two-dimensional graph (Figure 1).

The left graph of Figure 1 represents the combination of probabilities of answers to the two questions we selected. The red cross represents answers to Question 1 and the blue square represents answers to Question 2.

The right graph of Figure 1 presents the cumulative number and the volume of responses. Thus, in the mosaic graph, the largest areas represent Preferences 4 and 5. We also graphically confirmed the dependence between the investigated questions. The same is true for Options 3 (‘I can’t speak’) and 4 (‘I rather agree’). The only difference was in the choice of Options 1 (“strongly disagree”) and 2 (“rather disagree”). Respondents preferred Option 1 (“strongly disagree”) when choosing the answer to Question 1 and preferred Option 2 (“rather disagree”) when answering Question 2 (statement).However, there are very slight differences in preferences. The questions follow each other logically; therefore the confirmation of the results by correspondence analysis is a logical result. The results of the correspondence analysis show that this is a clear dependence, which has already been confirmed by the Chi-square test; the respondents, as representatives of companies, completely or at least partially agreed with the claims. Companies that follow a strategic approach to human resources in order to support innovative practices also implement innovative practices in the field of human resources. It follows that there is a consensus within companies that declare that they follow a strategic approach in the field of human resources in order to support innovative practices and at the same time implement innovative human resources practices.

We used second research assumption to find out whether companies implementing innovative practices in the field of human resources are also trying to create a quality working atmosphere and relationships with employees in order to meet the company’s goals. The research assumption was formulated as follows:

RA2: We assume that companies implementing innovative practices in the field of human resources also do everything they can to make employees feel engaged and dedicated to their work.

**Hypothesis** **0** **(H0).**
*There doesn’t exist a dependence between companies interest in implementation of innovative practices in the field of human resources and active management’s approach to creating conditions to make employees feel engaged and dedicated to their work.*


**Hypothesis** **1** **(H1).**
*There exists a dependence between management’s interest in the opinions and attitudes of their employees on the topic and active management’s approach to creating conditions to make employees feel engaged and dedicated to their work.*


Table 2 (above) is the result of the χ^2^ test, which verified the validity of the statistical hypothesis expressing the relationship between the statements.

Based on the value of probability, the value of which in all three tested interdependencies was less than 0.05. We can say that the results also confirmed the existence of a statistically significant dependence between management’s interest in the opinions and attitudes of their employees on the topic and active management’s approach to creating conditions to make employees feel engaged and dedicated to their work. We again identify the degree of dependence as moderately strong, since the observed value of the Cramer correlation V coefficient, which takes values from the range 0 to 1, is 0.5546.

We can interpret the relative position of the categories within one variable and at the same time their position relative to the categories of the other variable. We measure the degree of similarity based on Euclidean distances between points within one variable.

In Figure 2 we can see the uniformity in the attitudes of the respondents to both statements whose relationship we are examining. These are preferences for answers with a value of 4 and 5. This unambiguity of attitudes is no longer visible in other preferences.

The left picture in Figure 2 represents the combination of probabilities of answers to the two questions we selected. The red cross represents answers to Question 2, and blue square represents answers to Question 4. For example, Preference 3 (“I can’t answer”) was mentioned by fewer respondents in the context of “management’s interest in the views and attitudes of its employees” and by more respondents in favor of “active management approaches”. This creates a significant gap between the three values that were highlighted in the two selected statements of the questionnaire survey. In the mosaic graph (picture on the right side of Figure 2), where the largest areas of the graph represent Preferences 4 and 5, we can again graphically confirm the dependence between the examined questions, which has already been identified by the Chi-square test.

The last research assumption of this part of the work was based on the expectation that the company’s management is interested in the views and attitudes of employees to implement innovative practices in working with human resources, communicates with them, and employees have the opportunity to express their views and can subsequently perceive these changes and innovative practices positively.

RA 3: We assume that in companies whose management is interested in the opinions and attitudes of employees on the issue, according to the management, changes and innovations are perceived positively by employees.

**Hypothesis** **0** **(H0).**
*There doesn’t exist a dependence between company’s interest in the views and attitudes of their employees and employees’ perception of changes and innovations from the management point of view.*


**Hypothesis** **1** **(H1).**
*There exists a dependence between company’s interest in the views and attitudes of their employees and employees’ perception of changes and innovations from the management point of view.*


Based on the determined value of probability, which was even less than 0.05 in the last verified assumption, we can state that a statistically significant dependence was confirmed. This means that in companies whose management is interested in the opinions and attitudes of employees on the issue, according to management, changes and innovations are perceived positively by employees. The degree of dependence is also identified in this case as moderately strong, as the value of the Cramer’s correlation V coefficient is 0.5019.

On the left graph of Figure 3, it can be seen that the preferences of Values 4 and 5 are similarly indicated in both statements, which are part of the research assumption (RA3). Red cross represents answers to Question 5, and blue square represents answers to Question 3.

The preference of answer with a value of 3 appears to be remote due to the low number in the choice of answer, which is also declared by the mosaic chart. We can state that the values of Preferences 1, 2, and 3 differ considerably in the examined questions, while the unambiguity in the answers in the preferences is not obvious. Moreover, the most important finding is that the most common preferences reported by respondents are Options 4 (“rather agree”) and 5 (“strongly agree”). The largest areas of the graph on the right side of Figure 3 represent Preferences 4 and 5. We can again graphically confirm the dependence between the examined questions, which has already been identified by the Chi-square test.

## 4. Discussion

As we found out, companies that follow a strategic approach to human resources in order to support innovative practices also implement innovative practices in the field of human resources. It follows that there is a consensus within companies that declare that they follow a strategic approach in the field of human resources in order to support innovative practices and at the same time implement innovative human resources practices. Stachová and Stacho [24] recommend implementing specific processes aimed at realistically increasing employee involvement in finding and solving problems related to the company’s activities and the subsequent implementation of the submitted proposals. To ensure intensive involvement of employees in these processes, we recommend the company’s management declare to all employees their intention to implement such changes and to continue to support their activity in this direction in the form of reward or other motivation for each proposal, which will subsequently support their re-engagement in solving another problem. In this way, the potential to create a habit of employee involvement in innovation is supported.

Based on our findings on the companies whose management is interested in the opinions and attitudes of employees on the issue, according to management, changes and innovations are perceived positively by employees. Leaders play a key role in influencing employees to bring about organizational change. It is less known whether leaders who take advantage of transformational behavior will have a short-term and/or longer-term impact on employees’ attitudes to change. Results of the research made by Henricks et al. [42] showed that management errors at the beginning of a change can be corrected during implementation. The findings of Yilmaz et al. [43] suggest that the methods used, such as informing the employees about what is going on regarding change, consulting them, and maintaining participation of the employees in the change process, have a positive impact on the attitudes of managers towards change. This means that if an organization complies with the advice of the change literature asserting that the employees should be informed of, consulted, and participate in the change from the outset of the process, the employees would probably commit themselves to change rather than resist it. Shin et al. [44] also found that employees’ later commitment to change was a strong predictor of employees’ later behavioral support for change and turnover intention.

Our results confirm the existence of a statistically significant dependence between management’s interest in the views and attitudes of their employees on the topic and active management’s approach to creating conditions to make employees feel engaged and dedicated to their work. According to Šestáková and Hegedüs [30], employees can have a negative attitude, which can be overcome by continuous work, focused on communication and explaining the benefits of open innovation, as well as by looking for effective forms of motivating people. It is also important to encourage employees to exchange knowledge and build their relationship with the company, as well as to support the idea of exchanging information and knowledge between employees, organizing frequent meetings, and providing appropriate infrastructure for meetings, including virtual ones [15]. Kožárová and Sirotiaková [26] recommend considering the opposite behavior of employees. To avoid unwanted aversion or resistance, companies can choose appropriate strategies to reduce and overcome the negative effects of resistance to organizational change and the subsequent proper implementation of these strategies, which are essential for the successful implementation of organizational change.

Individual- and group-level openness to organizational change are important predictors of successful outcomes. Employees should be open both to the content of the change and to the process by which the intervention is implemented to maximize outcomes [45]. Onyeneke and Abe [46] found visioning, communication, participation, support, and concern for change participants’ interests to be of significant importance in ensuring employee buy-in and support for planned change efforts. Although change leadership had no direct effect on employees’ behavioral intentions to support change, it was strongly related to employee cognitive appraisal of change. The relationship between change leadership and employee behavioral intentions to support planned change was serially mediated by employee cognitive appraisal and emotional response toward the planned change event.

## 5. Conclusions

New procedures, the implementation of which is encouraged by managers, are more effective if employees are also actively involved. The performance also depends to a large extent on the application of new procedures in the field of human resources management. Innovative activities of companies do not have to be perceived only positively by employees. As part of the implementation of innovations and innovative procedures in working with human resources, it is appropriate to consider the opposite behavior of employees in order to minimize adverse events. Thus, it is recommended that companies formulate appropriate strategies to reduce and overcome the effects of negative attitudes to organizational change and then implement them appropriately, which is essential for the successful implementation of innovative practices. Communication is an essential part of such strategies. Promoting open corporate communication at all levels and in all directions, as well as the necessary feedback on employee performance and satisfaction, will support their loyalty and commitment. Creating opportunities to involve employees in decision-making, strengthening the position of employees with a higher degree of delegation, and supporting the autonomy and responsibility of employees requires competent managers as well as openness of companies to new challenges and procedures.

The limiting factor of the survey is the fact that the survey was not carried out in the ranks of executive employees, which would allow the implementation of comparative analysis. This is planned to be implemented shortly.

We recommend the company’s management actively involve employees in solving tasks related to the company’s activities within the practical implementation of innovative procedures, not only by comprehensible, clear, and appropriate level presentation of the intention to implement changes, but also by encouraging and motivating employees to be creative and present proposals, e.g., in the form of a reward for the completed proposal.

## Figures and Tables

**Figure 1 behavsci-12-00174-f001:**
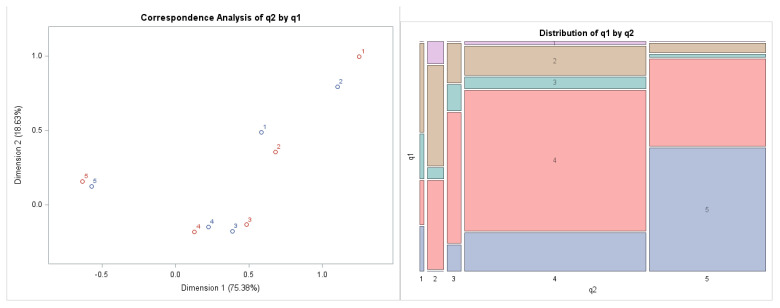
Outputs of the correspondence analysis: Correspondence Map and Mosaic Chart Combination between Questions 1 (Company follows a strategic approach to human resources in order to support innovative practices) and Question 2 (Company implements innovative practices in the field of human resources). Source: own data and own calculations.

**Figure 2 behavsci-12-00174-f002:**
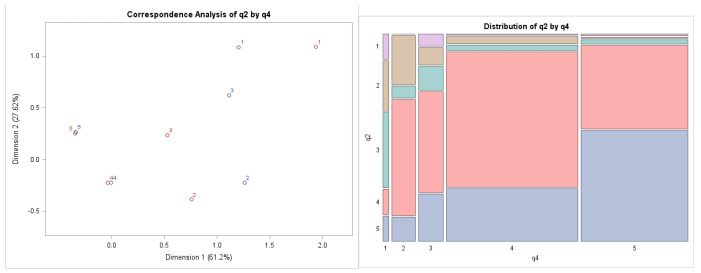
Outputs of the correspondence analysis: Correspondence Map and Mosaic Chart Combination between Question 2 (Company implements innovative practices in the field of human resources) and Question 4 (Management of the company does everything to make employees feel engaged and dedicated to their work). Source: own data and own calculations.

**Figure 3 behavsci-12-00174-f003:**
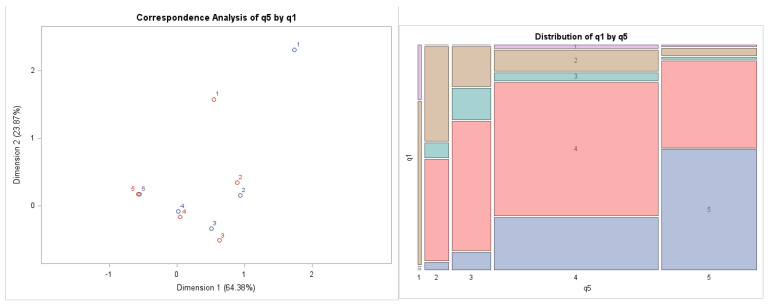
Outputs of the correspondence analysis: Correspondence Map and Mosaic Chart Combination between Question 5 (From the management of the company’s point of view, changes and innovations are perceived positively by employees) and Question 3 (Company is interested in the opinions and attitudes of their employees). Source: own data and own calculations.

**Table 1 behavsci-12-00174-t001:** Characteristics of the respondents according to identifications questions.

	Frequency	Percent
*Business size*		
Small business	248	30.39
Medium business	270	33.09
Big business	298	36.52
*The legal form of business*		
Joint-stock company	228	27.94
Cooperative	25	3.6
Other	40	4.90
Limited Liability Company	523	64.09
*Capital participation of the company*		
Combined equity participation	239	29.33
Exclusively domestic business	393	48.22
Exclusively foreign enterprise	183	22.45
*Industry*		
Primary sector	103	12.62
Secondary sector	331	40.56
Tertiary sector	332	40.69
Quaternary sector	50	6.13
*District of Slovakia*		
Bratislava region	186	22.82
Banská Bystrica region	64	7.85
Košice region	35	4.29
Nitra region	245	30.6
Prešov region	47	5.77
Trenčín region	73	8.96
Trnava region	104	12.76
Žilina region	59	7.24
operates throughout the Slovak Republic	2	0.25

Source: own data and own calculations.

**Table 2 behavsci-12-00174-t002:** Results of Chi-square test of the square contingency.

	Probability of Chi-Square Test	Cramer’s V
Research assumption 1	<0.0001	0.5786
Research assumption 2	<0.0001	0.5546
Research assumption 3	<0.0001	0.5019

Source: own data and own calculations.

## Data Availability

Data supporting the reported results can be accessed on request to the main author.

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
