# Peer review of "Attitudes and Involvement of Employees in the Process of Implementing Innovations and Changes in Companies"

_behavsci, 2022, doi:10.3390/bs12060174_

Round 1

Reviewer 1 Report

The article makes good contribution to the discourse of innovations infusion into organizations and the need for employee involvement. Its well written and structured. However, there are few sections requiring improvement as outlined below:

line 15-19: You have mentioned that you collected your quantitative data using a survey questionnaire and proposed hypotheses for the study. However, going deeper into the paper, i.e. line 179-185, your first 2 research assumptions should be structured properly like hypotheses and reflect whether the relationship is positively influencing the other.

Line 186 - 188: Your RA3 is not clear due to the double-barrelled relationship you have use, i.e. have a one-to-one/dependent-to-independent variable relationship, e.g. opinions and attitudes are two different things and can be bundled and measured as one variable, unless you clearly define it before, in terms of its construct items. Similarly, changes and innovations should be unbundled.  Maybe you may come up with two different hypotheses from RA3: Hypothesis (H3): Employee attitudes positively influence innovations in organizations etc.

Line 297- you have used Chi-Square Test to test your hypotheses! However, Chi-Square Test can only be used to test null hypotheses; yet in your case you hypotheses are not defined as null. Suggestion: (1). redefine your hypotheses as null hypotheses or (2). adopt a more rigor approach to testing hypotheses in-line with how you have defined them i.e. use confirmatory factor analysis using e.g. SmartPLS, AMOS, LISREL, EQS, SAS, etc.

Line 416: Please incorporate the changes made in the above section in your discussion.

Reviewer 2 Report

From the original stand point, the subject is not new. Employee participation in organizational change is a constant topic at least in Work and Organizational Psychology, which is muy field of expertise. However, the manuscript incorporates two interesting points:
  1. the link between employees' participation in change processes and HRM.
  2. the qualitative methodology applies to a large sample.
So this study complements extant research of change management and the role of HR departments on it.   The study is mainly descriptive, but although I'm not a methodologist, statistical operations seem robust to me. It is not a theoretical paper which could demand the explanation of the specific mechanisms through which employees decide to participate...   One of the conclusiones, which is that change processes will generate negative attitudes in employees, is very real.   Overall, I think the paper is well structured, clear, has two strong points (as mentioned) and it also has value for HR departments in order to strengthen their role in companies' change programs. 

Some formatting and citation suggestions:

In the theoretical section, please try and look for some more recent references.

Lines 31-32, use "people" instead of Human Resources.

Paragragh initiates in line 39, may be is better to talk about "increase their competitive advantages"

Line 45, before introducing the abbreviation, repite the whole words and them put the abbreviation between brackets.

52-53, I'm not a native speaker but past tense (emerged) with "during...". Please check

57, "...and IT needs..." Add IT as subject

67, "... remain committed to IT"

70, 71, 72, check "THE" before CHANGE. 

99-101, This affirmation requires citation.

111, remove "THE", begin with "ENGAGEMENT"

120, reference

125, THE

177, remove one space

476, Point before "WE". Begin new sentence with "THUS" or similar
